# Multimedia Interventions for Neurodiversity: Leveraging Insights from Developmental Cognitive Neuroscience to Build an Innovative Practice

**DOI:** 10.3390/brainsci12020147

**Published:** 2022-01-23

**Authors:** Teresa Farroni, Irene Valori, Laura Carnevali

**Affiliations:** 1Department of Developmental Psychology and Socialisation, University of Padova, 35131 Padova, Italy; irene.valori.1@phd.unipd.it (I.V.); laura.carnevali@phd.unipd.it (L.C.); 2Padova Neuroscience Center (PNC), University of Padova, 35129 Padova, Italy

**Keywords:** autism spectrum disorders, bodily self, multimedia interventions, technologies, virtual reality

## Abstract

Multimedia technologies and virtual reality offer unique possibilities to manipulate sensory, motor, interpersonal, and cognitive processes contributing to atypical developmental trajectories, thus holding an explosive potential to design innovative and engaging interventions. However, there has been little progress in developing interventions that go beyond the patient’s diagnosis or the fascination of technology and rather spring from a deep understanding of the specific neuropsychological processes to be nurtured in individuals. This perspective paper outlines how recent insights from developmental cognitive neuroscience can be leveraged to promote children’s multidimensional development and highlight future directions and challenges for innovating both research and clinical practice. Finally, we focus on some practical examples of multimedia and virtual reality activities we have designed to stimulate bodily-self experiences, which are crucial for building up a coherent sense of self and lay the foundation for interacting with the external world. Atypical bodily self is an early marker of heterogeneous neurodevelopmental conditions (such as autism spectrum disorders) and seems to be under-targeted in research and clinical approaches.

## 1. Introduction: Building an Innovative Practice

For many years, interventions for individuals with autism spectrum disorders (ASD) have been strictly related to the core symptoms of disease and therefore often proposed on this basis. This “diagnostic” approach is becoming less convincing, as it is clear that any specific deficits are not necessarily present in all individuals with ASD, not specific to ASD, and not sufficient to explain the heterogeneity of the spectrum [1,2]. Thus, time has come to go beyond labels and provide interventions based on specific functions and mechanisms rather than diagnosis per se. From being considered pathology-related impaired areas, functions are rather to be conceived as a starting point for an individualized intervention. A functional analysis in the observation of atypical development is proper of many recent approaches, yet a further step to be taken is to observe and unbundle each function into its components.

To address functions and leave diagnoses aside, we need to know how they are neuromodulated throughout development. According to Neuroconstructivism, development is a dynamic process that integrates individuals’ genetic predispositions and environmental experience in a probabilistic epigenesis [3]. The developing brain continually modifies its architecture and functioning according to the specific environment in which it is embedded. This is made possible by cortical connections that strengthen or weaken as a function of individuals’ activity (Interactive Specialization theory [4]). Importantly, our senses are the gateways through which a given stimulus can be perceived and processed to build up the individual phenotype. Multisensory processing becomes specialized with age such that the developmental stage determines what information needs to be processed and how and when the different inputs are combined [5]. The integration of multisensory information coming from inside and outside ones’ bodies also contributes to building the basis of the bodily self from the earliest stages of life, laying the foundation for interpersonal connections and environment exploration [6]. The variety of one’s sensory experience also changes as a function of motor development such that increasing motor skills allow individuals to navigate the environment exploring its different facets. Sensorimotor development thus appears the starting condition for increasingly complex functions that are observed over different developmental stages. In terms of intervention, stimulating low-level sensory processes—which are already known to come first in the timeline of brain development—and motor skills allows us to shape higher cognitive processes known to specialize later.

A promising tool that might be employed to both disentangle and intervene on specific functions are new technologies, which become the driver to change specific, atypical developmental pathways. In recent years, many authors have raised the question of whether digital technologies can be effectively used to intervene on core symptoms of ASD, arguing that they have the great potential of being easily delivered to a high number of individuals, adopting a motivating and fun approach [7,8,9]. Some efforts have been made to categorize previous studies on the use of technology in ASD intervention according to targeted skills (namely cognitive, adaptive, challenging behavior, joint attention, motor, school readiness, play skills, academic, communication, and social) [8]. However, literature approaching single functional components and technology-based intervention whilst going beyond the diagnostic label is poor. A further step yet to be taken consists in understanding which technologies are more appropriate or particularly promising to intervene on which dimension and in which instances and modalities.

Considering that new technologies allow us a great specificity of stimulation and intervention, in this perspective paper, we propose that to bridge research and clinical practice, it is necessary to use digital tools while looking at the function to be supported rather than at the actual diagnosis. Such a perspective not only encourages a new approach to intervention but also provides a novel framework for the underlying research choices and further opens a transversal reflection on neurodiversity. In our opinion, the challenge should be to address, in a systematic way, the act of steadily stepping up the level of intervention, starting from functions’ assessment until complete assistance is provided. Once users’ needs are precisely observed and outlined, and the helpful technologies are identified, a modular, suitable system should be developed to address the subjects’ strengths and weaknesses.

In the following sections, we address the pillars on which to build multimedia interventions: the senses, action and movement, social relationships, and cognitive skills. Finally, we apply these key concepts by illustrating some examples of multi-technology activities designed to stimulate the bodily self in children with neurodiversity.

## 2. Keep the Senses in Mind

Atypical perceptual and multisensory functioning greatly contribute to the profound heterogeneity that characterizes neurodevelopmental disorders and particularly those of the autism spectrum. From early ages and throughout the entire life cycle, individuals with ASD manifest difficulties and peculiarities in the processing and use of sensory information from both the external world (exteroception) and their own bodies (interoception and proprioception). More specifically, patterns of hypo- and hyper-responsiveness to sensory information are often found in ASD [10,11], along with atypical reliance on visual and proprioceptive information [12,13,14,15] and atypical brain responses to both affective and non-affective touch [16]. Crucially, the intertwining of multisensory information gives rise to an integrated bodily self-representation [17] that is crucial to interact with others such that impairments in these domains could contribute to socio-communicative difficulties observed in ASD [18]. Some studies suggest that body awareness in adults with ASD compared to controls’ is less reliant on body-related visual information [19]. This different experience of one’s own body is associated with the severity of ASD traits [20] and could be linked to difficulties with multisensory integration with potential negative effects on social skills [19]. For example, an increased reliance on body-based interoceptive cues impairs the use of external information, which is critical for interacting with people and objects around us [21]. The balance between processing and perceiving what is happening inside or outside the self is central to social cognition of understanding both the similarity and distinction between the self and others [22]. The cascading effects of sensory atypicalities on the subsequent development of higher-order abilities (e.g., executive functions and social communication) highlight the need for interventions (better if early intervention) that focus on stimulating multisensory processes.

In recent years, there has been mounting interest in the investigation of the potential that digital and multimedia technologies might have for sensory stimulation of people with ASD. Worldwide, some research teams employ multimedia interactive environments that provide sensorimotor stimulation, enable the communicative use of the children’s body, and foster their curiosity and sense of agency [7,23,24,25]. Mixed and immersive virtual reality as well as interactive multimedia environments can be developed and implemented to offer activities that provide individualized sensory stimulation suitable even for people with low cognitive and communicative functioning. For instance, one promising technology for multisensory stimulation for ASD is Immersive Virtual Reality (IVR) (for a review, see [26]). Specifically, immersive head-mounted displays (HMDs), compared to traditional monitors, have unique features that fully immerse the user in simulations of reality, increasing the sense of presence in the interaction space, allowing to provide and manipulate visual, auditory, tactile, vestibular, and proprioceptive information. The contribution of each individual sensory channel and multisensory integration to perception and movement can be assessed and stimulated based on the individual needs and functioning of the child and adult. Recent studies further suggested that headsets can stimulate multiple sensory systems in people with sensory processing disorders [27], promote multisensory integration in cases of vestibular disorders [28], and provide three-dimensional auditory stimuli in a play context, thereby reducing perceived anxiety toward target auditory stimuli in adolescents with ASD [29]. This corpus of studies indicates that stimulation via IVR might be relevant for people with sensory processing difficulties over and above the specific diagnostic label they were ascribed. As such, IVR appears to hold promise for the development and/or enhancement of socio-communicative skills rooted in low-level sensory aspects. In this context, the body is fundamental to situate in and experience a social and interactive context.

## 3. Promote Action and Motion

The way we perceive, learn, and explore both our internal and external world is particularly intertwined with motor development and skills [30]. Up to 87% of ASD population manifest motor impairments, which are reported to be comparable to cognitive and language impairments in terms of prevalence but are still underrecognized and rarely addressed during interventions [31]. Children with ASD show a variety of motor impairments in the domains of praxis and fine and gross motor skills [32], asymmetrical gait [33], and impaired postural stability up to adolescence and adulthood [34,35]. Of note, a bidirectional inter-reliance between motor and cognitive processes has been highlighted by previous literature [36], which suggests the relevance of intervention targeting the motor domain to strengthen cognitive function and vice versa.

Motor development is not a trivial acquisition of milestones but a complex self-organization challenge to integrate the mechanical part of the body with perceptions, thoughts, emotions, and their physiological underpinnings [37]. Locomotor experiences play a crucial role in the acquisition of interaction skills by providing learning opportunities during specific sensitive periods. By moving across their environment, children in fact learn to integrate their actions with sensory information, exploring the self and the others. Atypical motor behaviors might interfere with communicative development by altering one’s social and sensory experiences, contributing to difficulties in language and social communication, which are indeed commonly observed in many neurodevelopmental disorders [38]. In line with this view, a recent meta-analysis shows not only delayed and qualitatively different motor behaviors from the earliest stages of development in ASD but that these atypicalities are associated with later socio-communicative impairments [39].

Given the unbreakable link between the development of sensorimotor processes and higher-order functions, options for sensorimotor interventions need to be explored. IVR is one of the most promising technologies to enable users to move and navigate in space with one’s whole body while immersed in digital environments, thus stimulating body awareness, promoting gross and fine-motor rehabilitation, and enhancing eye-motor coordination in diverse contexts [40,41,42,43]. Our recent study highlights how head-mounted displays change the way children and adults use visual and proprioceptive information to move and locate themselves in space, with a strong reliance on vision in the neurotypical groups [15]. In addition, we investigated individual differences in how children and adults with ASD move and perceive their movements with different sensory information available [44]. In our sample, participants with greater motor accuracy when visual landmarks were available performed better in reality. On the other hand, participants who were facilitated to move when only proprioception was reliable (thus manifesting an atypical facilitation to move in darkness) showed higher accuracy in IVR compared to reality. While IVR mainly relies on visual stimulation, other multimedia technologies can be used to exploit different sensory modalities to train motor skills. For instance, virtual realities built upon motion platforms and acoustic tools might increase self-motion perception of visually impaired people [45], with broader implications for the use of sound to empower bodily perceptions. On this account, virtual realities may be particularly facilitating learning environments for children and adults with specific sensorimotor profiles.

## 4. Together Is Better

Sensory- and motor-based intervention acquire increasing relevance when applied in an interactive context, especially in the case of impairments in social communication. Evidence-based clinical approaches for ASD emphasize the importance of implementing interventions in small peer groups, which are particularly suited to the promotion of socio-communicative and cognitive skills as well as adaptive behaviors [46]. Peer-to-peer interactions should therefore be facilitated by following the children’s initiatives and interests while continuing to pursue each child’s sensorimotor, cognitive, and learning goals. The social and interactive facets of interventions could also protect from mental health issues by preventing loneliness. In fact, while feeling close to other people promotes well-being, feeling disconnected has been shown to compromise mental and physical health—in both neurotypical and clinical samples—thereby strengthening a feeling of neurocognitive isolation [47].

Over the past decades, immersive virtual reality has been widely used for enhancing communication and social skills in safe and controllable yet ecological contexts. Children with ASD exposed to such interventions have shown improvements in nonverbal communication, initiative, and social cognition [48]. In addition, technologies can offer to pairs or small groups of children innovative activities that are designed to create a cooperative environment in which they learn not only from the adult but especially from their playmate. Collaborative virtual environments (CVEs) can be used with children with ASD to enable several users to remotely interact with the environment at the same time. Each person is represented by their unique avatar, acting, moving and navigating the environment independently, thus communicating directly when they are close enough to another user’s avatar [49].

Beyond the use of IVR for remote peer interaction (i.e., individuals are working together on a shared task or activity, but are physically apart), in-person interaction enables multiple users to work on the same virtual activity while also sharing the real space. The latter option enriches the audio-visual interaction of bodily signals, such as interpersonal touch, which may promote a sense of presence and social connection, thus improving the affective information conveyed during virtual communication [50]. Compared to reality, IVR may offer unique possibilities to make the exposure gradual and adaptable to the individual while at the same time easily collecting data on the person’s behavior (choices made, type of exploration, up to including eye-tracking and kinematic aspects). This facilitates both research and intervention through the implementation of activities that respond in a predefined and controlled way to the user’s behavior.

In sum, individuals can make virtual experiences within realistic environments that can be programmed to manipulate sensory and social inputs at an optimal level for each individual. The role of caregivers, educators, and therapists in offering guidance and scaffolding to the child remain essential, as they must facilitate the experience through modeling, encouragement, suggestions, and reinforcement of target behaviors (Figure 1).

## 5. A Stairway to Cognition

From a neuroconstructivist perspective, working on low-level sensorimotor mechanisms has cascading effects on the stimulation of higher-level cognitive abilities. This is the prerequisite for impacting mechanisms such as memory, attention, and executive functions (EFs), which are commonly impaired in ASD and associated with behavioral and socio-communicative problems [51]. More specifically, EFs, defined as the set of skills that allows to regulate and control other cognitive functions and behavior to achieve a goal and adapt to new and complex situations, have been recently shown to mediate the association between sensory processing and behavior in ASD [52]. Being capable of self-regulation and flexibility indeed helps to better accept everyday situations, with repercussions on learning and adaptive behavior [53,54]. Moreover, executive processes are closely linked to socio-affective ones [55], thus benefiting from being trained in social situations.

Along with working on the sensorimotor and social domains to foster cognitive functions, directly strengthening attentive and executive processes appears crucial. Indeed, one of the main challenges for children with both typical and atypical development is “to learn how to learn”, which suggests EFs that are trainable and can be improved with practice [56]. To this end, research and interventions should go beyond the diagnostic label and identify an individual’s points of fragility and need for strengthening on the sub-components of each cognitive function. The interactive specialization approach [4,57] lets us predict that by working on a certain function, we will have a well-rounded change in both targeted and additional brain areas and neuropsychological functions that are interconnected. For example, emerging evidence indicates that selective attention is crucial for modulating activity of neural circuits involved in information processing. When presented with auditory stimuli and selectively focusing attention on phonological information, we activate a network that is not limited to the areas involved in auditory processing but also includes visual areas that are important for reading processes [58]. This suggests that stimulating selective attention could allow a rewiring of brain networks increasing their connectivity, potentially training multiple skills and processes at the same time.

For the observation of the child facing cognitive tasks of various types and the enhancement of their abilities, different interfaces and interactive tools can easily be customized with contents adapted to each user. Leveraging embodied cognition, we can repurpose classic cognitive tasks in a multisensory, motor, fun, and motivating version. Although virtual adaptations of well-known neuropsychological tasks are frequently used to capture neuropsychological constructs, there are contradictory findings about the correlation between performances in real and VR modalities [59]. For instance, IVR can be used to assess and train topographical memory with adapted versions of the Walking Corsi test, which seems to induce comparable performances in both real and virtual environments [60]. On the other side, an IVR gamified adaptation of the Wisconsin Card Sorting Test (tapping on cognitive flexibility), which allowed participants to not only match cards but rather navigate the environment and open doors [61], resulted in a poor correlation between performances in real and IVR modalities. This means that interactive modalities and contextual factors affect the neuropsychological functions we involve and measure with different technologies.

Expanding on this, some researchers suggested that the unique potential of VR is to go beyond the aforementioned construct-based tests of cognitive mechanisms, which may fail to predict functional behaviors in everyday situations and rather implement function-based assessments and interventions, whereby EF can be evaluated within ecologic and generalizable contexts that represent real-life tasks [59]. There is evidence to suggest that learning in a variety of social scenarios and environmental contexts promotes the generalization of skills trained in VR to everyday life situations [62]. For instance, adolescents with ASD who experienced IVR training on shopping skills showed, in a real supermarket, to be more capable than the control group in finding items and had greater confidence in the activity [63]. In this respect, IVR has also been used to support vocational training for adults with ASD and train skills, such as loading a truck, cleaning, managing money, and organizing shelves [64]. This approach would leverage executive processes in contexts similar to everyday activities that rely on those capabilities.

Importantly, learning should never be separated from fun. Indeed, optimal experiences are those leveraging enjoyment, which arises from going beyond what an individual has been programmed to do and achieving something unexpected. Enjoyment characterizes tasks where there is no concern for the self, goals are clear, and there is a reasonable chance of completion and immediate feedback. Moreover, people need to put forth a deep but effortless commitment and feel in control over their actions [65]. In this respect, the main benefits of virtual environments for individuals with ASD are that they are controllable and offer safe spaces for learning new skills in individualized situations [49]. They can be implemented to scale the level of predictability or uncertainty, thus pursuing a balance between safe harbor for the intolerance of uncertainty that often characterizes ASD [66] and the challenge of curiosity and flexibility that comes from being able to actively seek for new stimuli [67].

## 6. Practical Example: The Bodily Self

Imagine a young child named M. who refuses to see himself in the mirror or look at his image in family photos and videos. In case he accidentally happens to see himself, he frequently responds by screaming, self-harming, and attacking others. Do these surprising behaviors make more sense if you know he just received a diagnosis of ASD? How can caregivers “help” him face the anxiety of seeing his own image?

Now, picture another young child named A. who spends hours videotaping herself with a smartphone and pressing the play/pause command over and over again. While doing so, she produces excited vocalizations, grimaces, and hand flapping. Would you be surprised to know she also has a diagnosis of ASD? How do we interpret her behaviors towards her own image? She definitely looks happy but also repetitive in the way she restricts her interests, actions, and cognitive resources to the specific activity of watching her body.

We would probably believe that both children have an atypical body awareness despite their symptoms seeming almost the opposite. M.’s parents could simply start hiding the child’s photos and covering mirrors (a), while therapists might attempt to reduce those problematic behaviors through response extinction (b) or systematic desensitization techniques (c). Although such strategies might seem to work, relieving the child’s distress, reducing challenging behaviors, and/or making it possible for the child to actually see himself, it does not happen without a cost. Indeed, the child’s exposure to the self would end up being scarce (a), or the behavioral change would be likely neither be associated with modifications in the reasons driving the behavior (b) nor in its neurocognitive function (c). Interestingly, A.’s case might be even more challenging for parents and therapists. Why should they try to reduce a behavior that makes the girl satisfied and does not appear to be dangerous?

Although we can label M. and A.’s behaviors as part of general diagnostic features of ASD, such as hyper/hypo-reactions to sensory stimuli and repetitive and restricted behaviors, it would be more informative to focus on the underlying neurocognitive processes and wonder about their consequences on each child’s sensory, motor, social, and cognitive development. For instance, both children seem to have atypical body representations, which can be defined as “the ability to integrate multisensory (visual, proprioceptive, and tactile) bodily information into coherent representations of one’s own body” [68] (p. 1). The basic processes underlying body perception are now known to be present from birth, with newborns preferring synchronous rather than asynchronous visuo-tactile stimulation only when the visual stimulus has the typical configuration of infant bodies [69]. From the first months of life, babies engage with their reflection in a mirror and learn to move their bodies to make things happen. These precursors of later abilities of self-recognition, body ownership, and localization are not only fundamental for sensory and motor development but also the basis of intersubjectivity and social competence [70]. Notably, the fascinating question of how these multifaceted body representations gradually result in typical or atypical developmental profiles remains open but surely finds its roots in very early stages of the infant development.

A fascinating approach to stimulate these processes in children with ASD is to retrace, support, and scaffold the underlying developmental processes. We see in the box (Figure 2) some practical examples of the digitally aided activities we have designed in this direction over the course of six years of workshops with children and families. Our NUIEVE laboratory has been raised from the scientific, clinical, technical, and artistic collaboration of a multidisciplinary team of experts in the fields of developmental cognitive neuroscience, new multimedia technologies, visual arts, and music (https://www.facebook.com/NUIEVE/, accessed on 20 January 2022). Although ASD is one of the most frequent diagnoses received by children who have attended and attend our workshop, our activities have developed from the encounter with the uniqueness of interests, strengths, and weaknesses of each child. For this reason, and with the awareness of the great heterogeneity of children with neurodiversity, all activities are designed to be flexible and suitable for children with very different functioning profiles. We have interfaces designed for children with reduced mobility (i.e., enabling them to interact while sitting, laying on the floor), communication skills (i.e., providing opportunities to control the environment through movement and vocalizations), and cognitive abilities (leveraging procedural and intuitive learning that do not require understanding of verbal instructions). Each game leaves to the operators the possibility to change the contents (images, sounds, videos), the modes of presentation (size, speed, quantity, volume of the stimuli), and to create individualized and tailored paths.

These activities are suitable for both M. (for whom the bodily self seems fragile and a source of distress) and A. (for whom the bodily self appears a strong source of pleasure). Therapists can gradually expose M. to simplified, new, and curious body representations, going from a ghost-like shadow that mirrors his movements and produces congruent sounds to more realistic self-images that place him inside his favorite animations. From our anecdotal experience with children, this is a promising way to reduce avoidance and promote curious exploration of one’s body image, and it is also applicable to other avoided stimuli. On the other hand, therapists can exploit A.’s interest and motivation in exploring her body images to broaden her interests and train cognitive and social skills. For example, one can use her picture as a stimulus in activities tapping on memory (by constructing a memory game with pairs of different pictures of M.), sustained attention (by constructing an activity of finding her own picture among various distractors, along the lines of the Bells Test), and social imitation (having M. and a therapist or peer facing the digital mirror and trying to adopt identical postures). Again, from our anecdotal experience, this approach can boost children’s motivation, enjoyment, and self-efficacy, thus enhancing training possibilities.

## 7. Conclusions and Future Directions

The present work outlines key principles for developing innovative interventions for ASD, going beyond the diagnosis, and focusing on individual specifics to tailor stimulations that synergically work on low-level and high-level dimensions. We presented some examples of activities that leverage evidence from developmental cognitive neuroscience research (which considers the neural bases involved in each function) and use digital tools and virtual reality to promote change in the bodily self and related neuropsychological functions.

It is worth emphasizing that there are still major challenges to making this approach effective and applicable. At the research level, it will be critical to delve into the impact that each technology has on the neuropsychological functions of interest. The feasibility and generalizability of studies in this field is limited by the heterogeneity of developmental trajectories and the need for individualized interventions. It will be necessary to establish and test the optimal timing, frequency, and duration of these interventions to evaluate the effects of these simulations on children’s sensory, motor, cognitive, and social development and to assess their generalizability to daily life contexts and durability over time. Another important line for future research is to investigate the neural mechanisms underlying the different functions as they interact as well as the effect of training on the neural level. For instance, collecting neural measures before, during, and after an innovative training that uses digital technologies not only is crucial to deepen how the bodily self is expressed according to the various stimulations but also becomes particularly informative for clinicians.

In clinical practice, one of the main challenges is the multidisciplinary preparation of the team, which requires both psychological and technical expertise. A multimedia laboratory cannot rely only on commercially available tools that are often designed for gaming and not suitable for clinical needs. Consequently, there is a great deal of implementation and programming work not only initially but also in the interim for monitoring and readjusting tools and activities. Such experts and resources are often unavailable to clinical settings, and they constitute substantial costs that limit feasibility and affordability. Finally, communication and alliance with families is fundamental. We should not only take care of their requests for the well-being of the child and whole family but also clearly illustrate goals and methods that may be unfamiliar. Moreover, it is ethical to communicate them the experimental nature of scientifically newborn methods and to make them co-constructors of the pathway.

Although the road is long and uphill, we believe that this perspective paper can contribute to a shared reflection on these topical issues, leading research and clinics to make synergistic steps towards new and promising approaches to understand and treat neurodevelopmental disorders, such as ASD.

## Figures and Tables

**Figure 1 brainsci-12-00147-f001:**
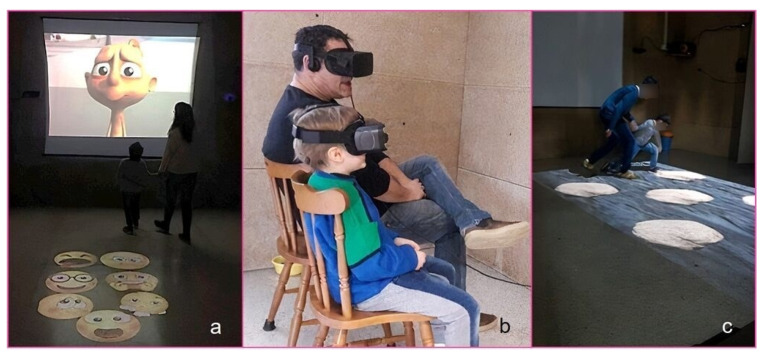
(From left to right) Therapists guiding children through (**a**) emotion recognition activity, (**b**) immersive virtual exposure to daily life abilities (i.e., taking the bus), and (**c**) adaptation of a Walking Corsi test for visuo-spatial working memory.

**Figure 2 brainsci-12-00147-f002:**
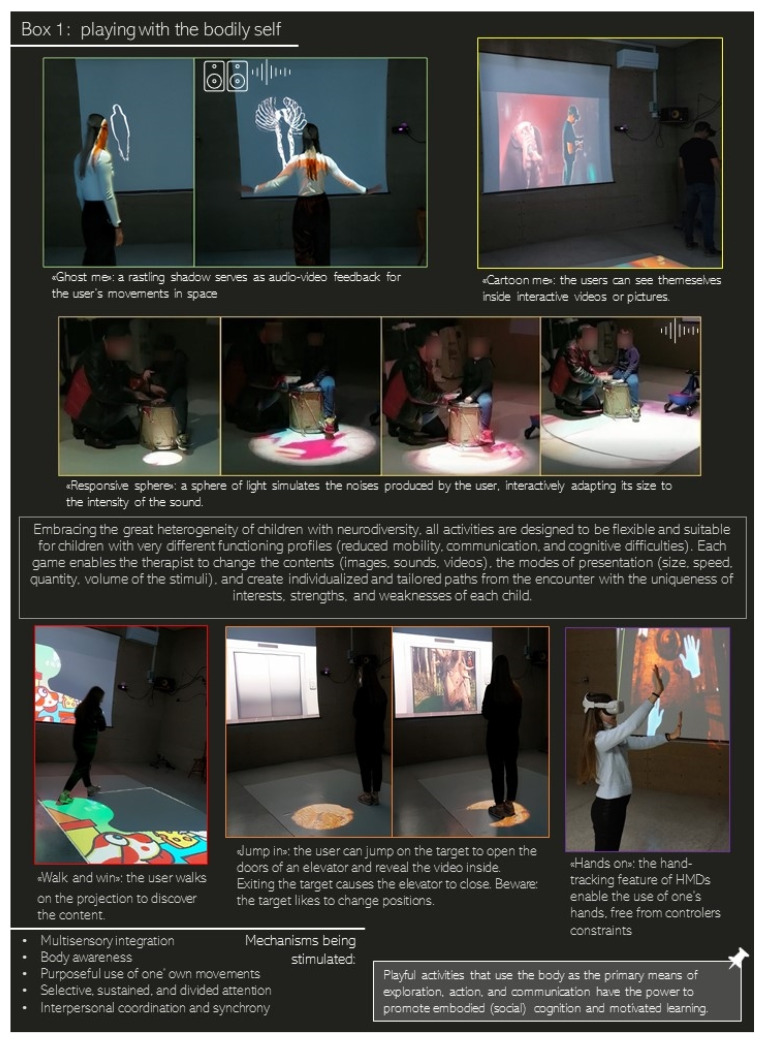
Examples of multimedia- and IVR-gamified activities to simultaneously foster sensorimotor, social, and cognitive mechanisms contributing to the bodily-self experience.

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
