# Peer review of "Multimedia Interventions for Neurodiversity: Leveraging Insights from Developmental Cognitive Neuroscience to Build an Innovative Practice"

_brainsci, 2022, doi:10.3390/brainsci12020147_

Round 1

Reviewer 1 Report

Thank you for asking me to review this excellent perspective article titled: "Future directions to bridge research with practice: insights from developmental cognitive neuroscience can be leveraged to promote multimedia interventions". The article focuses on how recent insights from developmental cognitive neuroscience can be leveraged to promote children’s multidimensional development, and highlight future directions for innovating both research and clinical practice. I can clearly see the good matching of the topic of the article with the Special Issue on "Epidemiology of ASD Services: Unmet Need, Barriers and Innovative Solutions".

The title is very long, and I wonder if the first part before the ":" (Future directions to bridge research with practice:) is really necessary. I would also give more evidence to the multimedia interventions, that in the final analysis is the really crucial/central part of the article (and of the Special Issue)

The authors decided to keep a very narrative approach, and their sections have very interesting titles. Although I find the decision very agreeable, I do suggest that for some clarity, the authors briefly describe at the end of the introduction (section 1) what is going to wait for the audience in the next sections.

In general, the authors present the case of the advantage of using a specific approach to the treatment of ASD, However, it would be interesting if some emphasis would also be given to the challenges. I think this would make the text stronger and more informative for the readers.

Finally, I noticed some typos, and some of the sentences look a bit too wordy. I encourage the authors to proof check the text again.

Author Response

Reviewer 1

The title is very long, and I wonder if the first part before the ":" (Future directions to bridge research with practice:) is really necessary. I would also give more evidence to the multimedia interventions, that in the final analysis is the really crucial/central part of the article (and of the Special Issue)

A: We carefully considered and agreed with the reviewer’s comment, and therefore changed the title accordingly. The new title states:

Multimedia interventions for neurodiversity: leveraging insights from developmental cognitive neuroscience to build an innovative practice”

The authors decided to keep a very narrative approach, and their sections have very interesting titles. Although I find the decision very agreeable, I do suggest that for some clarity, the authors briefly describe at the end of the introduction (section 1) what is going to wait for the audience in the next sections.

A: We appreciate this suggestion and included an introductive overview of the following sections:

In the following sections, we will address the pillars on which to build multimedia interventions: the senses, action and movement, social relationships, and cognitive skills. Finally, we will apply these key concepts by illustrating some examples of multi-technology activities designed to stimulate the bodily self in children with neurodiversity.”

In general, the authors present the case of the advantage of using a specific approach to the treatment of ASD, However, it would be interesting if some emphasis would also be given to the challenges. I think this would make the text stronger and more informative for the readers.

A: Thanks to the reviewer for pointing that out and giving us the chance to further discuss the ongoing challenges we see in the use of multimedia technologies for ASD. Please see the “Conclusions and future directions” section, which is now much expanded.

Finally, I noticed some typos, and some of the sentences look a bit too wordy. I encourage the authors to proof check the text again.

A: We proof checked the manuscript and fixed typos and long sentences.

Reviewer 2 Report

This paper reviews current multimedia interventions designed to foster ASD individuals' social and cognitive skill development. The overall narrative looks good to me. A suggestion is that you may want to use some words to highlight that it is a conceptual review paper that does not contain any methodology. Or if you want to clarify the methodology you used, I want you to add the method section that briefly introduces what methods you took for the paper. 

Author Response

Reviewer 2

This paper reviews current multimedia interventions designed to foster ASD individuals' social and cognitive skill development. The overall narrative looks good to me. A suggestion is that you may want to use some words to highlight that it is a conceptual review paper that does not contain any methodology. Or if you want to clarify the methodology you used, I want you to add the method section that briefly introduces what methods you took for the paper. 

A: As we agree on the narrative and perspective purpose of the present manuscript, we frequently refer to our work as such. The definition “perspective paper” is used

  • in the first page:

Perspective paper for Special Issue "Epidemiology of ASD Services: Unmet Need, Barriers and Innovative Solutions"

  • at the end of “Conclusions and future directions” section

“we believe that this perspective paper can contribute to a shared reflection on these topical issues […]”

  • Now further remarked in the introduction

“In this perspective paper we propose […]”

  • And abstract

This perspective paper outlines […]